# Patient-Derived Organoid Models for NKT Cell-Based Cancer Immunotherapy

**DOI:** 10.3390/cancers17030406

**Published:** 2025-01-26

**Authors:** Pablo A. Palacios, Iván Flores, Lucas Cereceda, Francisco F. Otero, Marioly Müller, Priscilla Brebi, Héctor R. Contreras, Leandro J. Carreño

**Affiliations:** 1Millennium Institute on Immunology and Immunotherapy, Programa de Inmunología, Instituto de Ciencias Biomédicas, Facultad de Medicina, Universidad de Chile, Santiago 8380453, Chile; 2Department of Basic and Clinical Oncology, Faculty of Medicine, Universidad de Chile, Santiago 8350499, Chile; 3Departamento de Tecnología Médica, Facultad de Medicina, Universidad de Chile, Santiago 8380453, Chile; 4Millennium Institute on Immunology and Immunotherapy, Laboratory of Integrative Biology (LIBi), Centro de Excelencia en Medicina Traslacional (CEMT), Scientific and Technological Bioresource Nucleus (BIOREN), Universidad de La Frontera, Temuco 4811230, Chile; 5Biomedical Research Consortium (BMRC), Santiago 8331150, Chile; 6Center for Cancer Prevention and Control (CECAN), Santiago 8350499, Chile

**Keywords:** tumor microenvironment (TME), 3D cancer models, patient-derived organoid (PDO), invariant Natural Killer T cells (iNKT), immunotherapy

## Abstract

Patient-derived organoids (PDOs) have revolutionized cancer research by offering three-dimensional models that retain the genetic and architectural features of primary tumors. The integration of invariant Natural Killer T (iNKT) cells into PDO platforms presents an innovative approach for studying immune–tumor dynamics and advancing cancer immunotherapy. iNKT cells, known for their ability to modulate immune responses and target tumors, face challenges in suppressive tumor microenvironments (TMEs), which limit their therapeutic potential. PDOs provide a physiologically relevant platform to explore strategies to counteract TME-induced immunosuppression, such as modulating tumor-associated macrophages (TAMs), myeloid-derived suppressor cells (MDSCs), or tumor metabolism. Additionally, PDOs enable the testing of combination therapies, including iNKT cell-based treatments and immune checkpoint inhibitors, to amplify anti-tumor efficacy. While the integration of iNKT cells with PDOs remains underexplored, ongoing advancements in multi-omics, imaging, and automation technologies promise to enhance this approach, paving the way for personalized cancer treatments.

## 1. Introduction

Cancer remains a leading global health challenge, accounting for millions of deaths annually and imposing significant socio-economic burdens worldwide [1,2]. The heterogeneity of cancer, both within and across tumor types, complicates treatment strategies and underscores the need for innovative therapeutic approaches [3,4]. Traditional treatment modalities, such as surgery, chemotherapy, and radiation therapy, have been instrumental in extending survival and achieving remission for many patients. However, their efficacy is often limited by their inability to target the underlying complexity of the tumor microenvironment (TME) [5,6]. The TME, composed of stromal cells, immune cells, extracellular matrix components, and signaling molecules, plays a critical role in mediating therapy resistance. For instance, stromal cells can provide survival signals to cancer cells, while immunosuppressive elements within the TME, such as regulatory T cells and myeloid-derived suppressor cells (MDSCs), can shield tumors from immune-mediated destruction [7,8,9].

In recent years, immunotherapy has revolutionized cancer treatment by harnessing the immune system to identify and destroy malignant cells. Among immunotherapeutic strategies, checkpoint inhibitors, CAR-T cell therapies, and cytokine therapies have gained prominence [10,11]. However, these approaches face challenges such as off-target effects, limited efficacy in immunosuppressive TMEs, and variability in patient responses [12,13]. One promising yet underexplored avenue is the application of invariant Natural Killer T (iNKT) cells in cancer immunotherapy. These specialized immune cells straddle the boundary between innate and adaptive immunity, enabling them to mount rapid and robust anti-tumor responses [14,15].

iNKT cells are unique in their ability to recognize glycolipid antigens presented by CD1d molecules on the surface of tumor and antigen-presenting cells [16,17,18]. Once activated, iNKT cells secrete a diverse array of cytokines, including IFN-γ and IL-4, among others, which not only mediate direct tumor cytotoxicity but also enhance the recruitment and activation of other immune cells, such as dendritic cells (DCs) and CD8^+^ T cells [19,20,21,22]. Despite their potential, the clinical application of iNKT cells has been hindered by challenges in their isolation, expansion, and functional optimization, as well as the suppressive influence of the TME, characterized by regulatory T cells (Tregs), myeloid-derived suppressor cells (MDSCs), and inhibitory cytokines such as TGF-β [11,23].

To overcome these challenges, researchers have turned to patient-derived organoids (PDOs) as advanced preclinical models for studying cancer biology and testing therapeutic interventions [24]. PDOs are three-dimensional structures cultured from tumor samples obtained directly from patients [25]. These models retain the genetic, histological, and functional properties of the original tumor, making them invaluable tools for personalized medicine [26,27]. Unlike traditional 2D cell cultures, PDOs replicate the spatial organization and cellular heterogeneity of tumors, providing a more accurate representation of the TME [5,28,29,30].

The integration of iNKT cells into PDO systems offers a novel platform to investigate tumor–immune interactions in a patient-specific context. By co-culturing PDOs with autologous immune cells, researchers may study the dynamics of iNKT cell activation, cytokine secretion, and tumor killing [14,31]. Furthermore, PDO models enable the exploration of strategies to overcome immune suppression, such as modulating CD1d expression or combining iNKT cell therapies with immune checkpoint inhibitors [14,32,33]. This approach has the potential to advance our understanding of iNKT cell biology and accelerate the development of effective cancer immunotherapies.

This review aims to provide a comprehensive overview of the current state of PDO research and its integration with iNKT cell-based therapies. We discuss the advantages of PDOs over traditional models, the unique properties of iNKT cells, and the challenges and innovations in combining these two technologies. By leveraging the strengths of PDOs and iNKT cells, we can address critical gaps in cancer research and pave the way for more effective and personalized treatment strategies.

## 2. Refining Tumor Models: PDOs Versus Traditional Systems

### 2.1. Limitations of Traditional Models

Traditional models, such as 2D cell cultures and animal systems, have been instrumental in advancing cancer research but are increasingly recognized for their inherent limitations. Two-dimensional cell cultures fail to replicate the three-dimensional architecture, heterogeneity, and cell–cell interactions that characterize the TME [30,34,35,36]. The absence of these critical elements often results in misleading data on tumor behavior and drug responses [28,37]. For instance, therapeutic agents that exhibit efficacy in 2D cultures frequently fail in clinical trials due to their inability to target the complexities of the TME [38]. This limitation underscores the need for models that better reflect the in vivo environment.

Animal models, while addressing some of the limitations of 2D cultures, present a different set of challenges. Differences in genetics, immune system architectures, and metabolic pathways between humans and commonly used animal models, such as mice, often result in poor translational accuracy [39,40]. Drugs that show promise in animal studies may fail to exhibit similar efficacy in human clinical trials due to species–specific differences in tumor biology and immune responses [41]. Additionally, the ethical concerns, high costs, and long development timelines associated with animal studies further limit their utility. These challenges emphasize the need for more accurate, efficient, and ethical preclinical models [29,42].

### 2.2. Advantages and Limitations of PDOs

PDOs have emerged as a transformative tool in cancer research by retaining the genetic, phenotypic, and architectural characteristics of the original tumors [24]. These three-dimensional structures, derived from patient tumor samples, enable researchers to study tumor biology and treatment responses in a context that closely mimics the in vivo environment [43,44]. One of the most significant strengths of PDOs lies in their ability to maintain tumor heterogeneity, including the presence of cancer stem cells, which are crucial for understanding tumor progression and therapy resistance [43,45]. Furthermore, PDOs support co-cultures with stromal and immune components, offering a comprehensive platform to study the dynamic interactions within the TME [46,47]. This capability is invaluable for evaluating immunotherapies, as it allows for the integration of autologous immune cells, such as tumor-infiltrating lymphocytes (TILs) and peripheral blood mononuclear cells (PBMCs), enabling real-time observation of immune cell infiltration, cytokine secretion, and tumor–immune dynamics [48,49]. Additionally, advances such as organoid biobanking and single-cell RNA sequencing have enhanced the scalability of PDOs, facilitating high-throughput drug screening and personalized medicine applications [50].

One of the key advantages of PDOs is their adaptability for high-throughput applications. Techniques such as CRISPR-Cas9 gene editing and organoid-on-a-chip platforms have further expanded the functionality of PDOs by enabling precise genetic manipulation and dynamic modeling of tumor–immune interactions [45,51]. These advancements allow researchers to explore specific genetic drivers of cancer and their interactions with therapeutic agents, thereby accelerating drug discovery and biomarker identification. Furthermore, PDOs offer unparalleled opportunities to study rare or treatment-resistant cancers, providing insights into tumor evolution and therapy resistance [31,52].

Despite their transformative potential, PDOs face several notable limitations that hinder their broader adoption. One challenge is the extended time required to establish viable organoids, which can take weeks or months, delaying experimental timelines and decision-making processes in clinical settings [31,53]. This is compounded by the resource-intensive nature of PDO culture systems, which rely on expensive specialized media, growth factors, and extracellular matrices, such as Matrigel, making them less accessible [50,54]. Another key limitation lies in replicating the TME with functional immune systems. While advances in co-culture techniques have improved TME representation, the integration of stromal and immune components remains technically challenging [43,48]. Moreover, prolonged culture conditions can lead to genetic drift and a loss of tumor heterogeneity, reducing the fidelity of PDOs in replicating patient tumors [45,55]. Achieving consistency and standardization across large-scale studies also presents significant hurdles, limiting their widespread adoption as universal preclinical models [51,52].

To address these challenges, ongoing innovations are reshaping PDO technology to enhance its scalability, fidelity, and functionality. Emerging strategies, such as the development of synthetic extracellular matrices and optimized biobanking protocols, are improving the accessibility and reproducibility of PDO systems [54]. Moreover, advancements in automated culture platforms and high-throughput screening technologies are reducing the time and cost associated with PDO generation, enabling their integration into large-scale studies [45,51]. Innovations like microfluidic systems and air–liquid interface cultures are also advancing the ability to incorporate stromal and immune components, providing a more comprehensive representation of the TME [43,48].

By addressing these limitations, PDOs are poised to play an increasingly central role in advancing cancer research and personalized medicine, offering a unique blend of complexity, scalability, and relevance that bridges the gap between preclinical and clinical studies.

### 2.3. Tissue-Specific Characteristics of PDOs

The tissue of origin significantly influences the characteristics, applications, and challenges associated with PDOs. Different types of tumors require tailored culture conditions and present unique opportunities for research and therapy development. Efforts to address the specific limitations associated with each tissue type are advancing the utility and translational relevance of PDOs (Table 1).

**Gastrointestinal (GI) Cancers**. PDOs derived from colorectal, gastric, and pancreatic cancers have been extensively used to study tumor–stroma interactions, therapy resistance, and immune modulation. These models are particularly valuable for assessing therapies targeting the Wnt signaling pathway, a key driver of tumor growth in GI cancers. Challenges in this domain include the incorporation of fibroblasts, endothelial cells, and immune components to better mimic the stromal and immune aspects of the TME. Additionally, achieving long-term culture stability while maintaining genetic fidelity and tumor heterogeneity remains difficult, especially in pancreatic cancer PDOs, which are prone to cellular senescence and loss of functional characteristics. Advances in co-culture techniques, synthetic stromal matrices, and optimized media formulations are addressing some of these limitations [4,43,56,57].

**Breast Cancer**. Breast PDOs represent the phenotypic diversity of subtypes, including hormone receptor-positive, HER2-enriched, and triple-negative breast cancer (TNBC). These PDOs are effective for testing immune checkpoint inhibitors and PARP inhibitors, particularly in TNBC, where therapeutic options are limited. However, the replication of dense extracellular matrix (ECM) and vascular structures, critical to breast cancer progression and therapy resistance, poses significant challenges. Additionally, breast PDOs often struggle to replicate the interaction between tumor cells and adipose tissue, a key component of the breast TME. Recent advances in hydrogel-based matrices, 3D bioprinting, and the incorporation of adipocytes and fibroblasts into PDO systems have shown promise in overcoming these limitations [58,59,60,61,62].

**Lung Cancer**. PDOs derived from lung cancers enable the evaluation of targeted therapies for driver mutations such as EGFR, ALK, and MET. These models are particularly useful for studying the interplay between hypoxia and immune evasion mechanisms. Challenges include maintaining long-term cultures while preserving the genetic integrity of highly metastatic subtypes, as well as accurately replicating the hypoxic conditions typical of lung tumors. Additionally, co-culturing lung PDOs with immune cells such as alveolar macrophages and T cells to mimic the immune microenvironment remains an area of ongoing research. Continuous optimization of culture media and genetic monitoring, as well as advances in hypoxia chambers and microfluidic technologies, are helping to address these issues [47,63].

**Liver Cancer**. PDOs from hepatocellular carcinoma (HCC) are instrumental in evaluating tyrosine kinase inhibitors, immunotherapies, and viral oncogenesis pathways. They are also used to explore the effects of viral infections, such as hepatitis B and C, on tumor progression. However, genetic drift and loss of tumor heterogeneity during extended culture are significant barriers, particularly in the study of viral dynamics. Another challenge is the lack of effective integration of stellate cells and immune cells, which are crucial for modeling the fibrotic and immune-driven aspects of HCC progression. Improvements in cryopreservation techniques, co-culture systems, and shorter culture timelines are helping to mitigate these issues [64,65].

**Prostate Cancer**. Prostate cancer PDOs provide critical insights into androgen receptor signaling and resistance mechanisms. These models are utilized for testing androgen deprivation therapies and next-generation anti-androgens. A major challenge lies in the limited availability of high-quality tissue samples, particularly from advanced prostate cancers, and replicating the prostate’s stromal environment. Current efforts focus on integrating cancer-associated fibroblasts (CAFs), immune cells, and nerve cells into prostate PDOs to better model the intricate tumor–stroma interactions. Additionally, advances in organoid biobanking are improving the availability and diversity of prostate cancer PDOs [66,67,68].

**Ovarian Cancer**. High-grade serous ovarian carcinoma PDOs facilitate the study of platinum resistance mechanisms and the efficacy of PARP inhibitors. These models are particularly useful for understanding immune evasion in ascitic environments, which are common in advanced ovarian cancer. Challenges include maintaining the complex ascitic microenvironment, which contains a mix of cancer cells, stromal cells, and immune cells, and ensuring genetic fidelity during long-term cultures. Efforts to address these challenges involve integrating immune components and developing ascites-mimicking matrices that better replicate the ovarian TME [54,69].

**Brain Tumors**. PDOs from glioblastoma and other brain tumors provide platforms for studying the blood–brain barrier’s impact on drug delivery and resistance mechanisms. These PDOs are increasingly used for testing CAR-T cell therapies and other immune-based treatments. However, challenges include replicating the neural microenvironment, maintaining the unique metabolic conditions of brain tumors, and incorporating functional vasculature into the models. Innovations in neural co-culture systems, bioprinting techniques, and metabolic profiling are being developed to address these issues [41,70].

**Table 1 cancers-17-00406-t001:** Application of PDOs in tumor studies.

Type of Cancer	Characteristics	Year	Main Discoveries	Type of PDO	References
Colorectal Cancer	Microsatellite stability (MSS) vs. instability (MSI)	2024	Identified cancer-specific tissue markers predicting ICI response; insights into immunotherapy resistance; gene-editing strategies.	3D PDO with immune system interaction	Esposito et al., 2024; Mo et al., 2022; Kim et al., 2022 [65,71,72]
Ovarian Cancer	Tumor heterogeneity	2023	High fidelity in molecular properties; tested drug sensitivity; analyzed ECM and TME for PDO relevance.	3D Dynamic PDO	Spagnol et al., 2023 [54]
Pancreatic Cancer	Tumor–immune interaction	2023	Modeled immune cell population changes using PDO-PBMC co-culture; explored personalized therapy strategies.	PDO co-cultured with PBMC	Knoblauch et al., 2023 [73]
Gastrointestinal Cancer	High cancer mortality	2024	Reviewed advancements in PDO drug screening and personalized medicine; TME modeling with ECM and ALI cultures.	3D PDO	Yang et al., 2024 [56]
Breast and Lung Cancers	Tumor infiltration and immune evasion	2024	Investigated tumor responses to immune cell therapies; TME modulation with cytokines and stromal components.	PDO with cytokine-modulated environments	Tong et al., 2024; Ding et al., 2022; Yokota et al., 2021 [26,47,63]
Lung Cancer	Immune evasion mechanisms	2021	PDO co-cultures reveal PD-1 blockade efficacy and immune cell recruitment.	Immune-enhanced PDOs	Xu et al., 2021 [34]
Esophageal Cancer	Drug resistance and progression under hypoxia	2022	Identified hypoxia-induced pathways contributing to resistance mechanisms.	Hypoxia-modified 3D PDOs	Zhou et al., 2022 [43]
HNSCC	Stromal interaction and immune suppression	2022	PDOs show robust immune interactions, supporting CD8^+^ T cell activation.	Immune co-culture PDOs	Kim et al., 2022 [72]
Cholangiocarcinoma	High intra-tumoral heterogeneity	2023	Characterized bile acid’s role in tumor progression using PDOs.	PDOs with metabolic modeling	Esser et al., 2020 [45]
Renal Cell Carcinoma	Microenvironmental effects	2020	PDOs predict response to VEGF inhibitors in renal cancers.	Matrigel-based cultures	Na et al., 2020 [74]

In conclusion, the tissue of origin plays a pivotal role in shaping the design, application, and translational success of PDO models. By addressing tissue-specific challenges, such as replicating microenvironmental complexity and maintaining genetic fidelity, researchers can enhance the relevance and utility of PDOs. Innovations in co-culture systems, bioprinting, and advanced culture techniques are driving the development of PDOs as versatile platforms for studying cancer biology and testing personalized therapies. These advancements are paving the way for more precise and effective approaches to cancer treatment.

## 3. Integrating iNKT Cells with PDOs

### 3.1. iNKT Cell Functions in Cancer and Immunotherapy

iNKT cells are a distinct subset of T cells that exhibit characteristics bridging the innate and adaptive immune systems. These cells express a semi-invariant T-cell receptor (TCR) that recognizes lipid antigens presented by CD1d, a non-polymorphic MHC class I-like molecule [18,75,76]. iNKT cells play a central role in anti-tumor immunity by modulating TME through cytokine secretion, direct cytotoxicity, and interaction with other immune cells [77,78]. Upon activation, iNKT cells secrete a wide array of cytokines, including IFN-γ, IL-4, IL-10, and TNF-α, which drive Th1 and Th2 immune responses [79,80,81]. These cytokines enhance the recruitment and activation of CD8^+^ T cells, DCs, and Natural Killer (NK) cells, amplifying anti-tumor immunity [19,20,82].

Interestingly, it has been reported that iNKT cells are altered in patients with cancer, and their absence can be linked to adverse outcomes, whereas their presence can be associated with beneficial outcomes. A study found that the response in patients with head and neck squamous cell carcinoma after radiotherapy was associated with the number of circulating iNKT cells. Patients with lower numbers (fewer than 48 iNKT cells per 10^6^ T cells) had the worst outcomes, while those with higher numbers had better outcomes [83]. Another study found that activated Vα24^+^Vβ11^+^ iNKT cells infiltrated tumors in patients with colorectal cancer. Notably, patients with higher levels of tumor-infiltrating iNKT cells had significantly better survival rates than patients with lower levels of iNKT cell infiltration [84].

The potential of iNKT cells in immunotherapy has also been explored. A phase I clinical trial found that α-GalCer administration in patients with solid tumors was well tolerated. However, the authors did not observe a clinical response since only seven patients were stabilized, and fifteen patients continued with tumor progression. The authors postulate that the low number of iNKT cells before treatment influenced the poor outcome of α-GalCer administration [85]. Another phase I/II clinical trial used PBMCs cultured with IL-2/GM-CSF and loaded with α-GalCer to treat patients with refractory or advanced non-small cell lung cancer. The study found that treatment increased the number of Vα24^+^Vβ11^+^ NKT cells in six out of seventeen patients. Also, PBMCs isolated from treated patients were restimulated with α-GalCer, and an increase in the IFN-γ-producing cells was observed in ten patients. Notably, those ten patients who exhibited higher IFN-γ-production had a better survival rate than those with lower IFN-γ production [86]. A study conducted by Exley and colleagues involved isolating iNKT cells from patients with advanced melanoma (stage III or IV) and expanding them ex vivo. The expanded iNKT cells were then administered to the patients. The authors concluded that the adoptive transfer of iNKT cells demonstrated good tolerance and led to immune cell activation in some patients, including CD4+ T cells, NK cells, and monocytes [87].

Despite their potent anti-tumor effects, iNKT cells often face suppression within the TME, probably due to Tregs, MDSCs, and inhibitory cytokines like TGF-β, which may impair their function and reduce their frequency [11,88]. This suppression contributes to the dysfunction of iNKT cells, which is associated with poorer outcomes in cancer patients [11,89]. Notably, the presence of iNKT cells correlates with better prognoses in several cancers, highlighting their therapeutic potential. Given that iNKT cells have been tested in clinical trials with varying levels of success, there is a pressing need for advanced tools to study and predict their responses in cancer. In this context, PDO platforms offer an ideal model to investigate the suppressive mechanisms of the TME, the transactivation of additional immune cells, and the direct cytotoxicity of iNKT cells. Furthermore, PDO platforms may aid in the development of strategies to reinvigorate iNKT cell activity and potentially overcome the barriers hindering the clinical translation of iNKT cell-based immunotherapies [88,89].

### 3.2. PDO Platforms for iNKT Research

PDOs represent an advanced preclinical model that faithfully recapitulates the architecture, genetic profile, and functional behavior of primary tumors. While iNKT cells have not yet been directly studied in PDO platforms, other immune cells, such as T cells, NK cells, and tumor-associated macrophages, have been successfully integrated into PDO systems to investigate tumor–immune interactions and evaluate immunotherapeutic strategies [26,90,91]. PDO–iNKT systems offer several unique applications (Figure 1).

#### 3.2.1. Evaluation of Th1-Biased α-GalCer Analogs in PDOs for Immunotherapeutic Potential

The immunomodulatory role of iNKT cells is rooted in their ability to produce T helper 1 (Th1)- and T helper 2 (Th2)-associated cytokines upon activation with α-GalCer. This dual functionality underpins the application of glycolipid ligands across various immune contexts, including cancer [81]. However, in this pathology, inducing a Th1-skewed response—characterized by higher IFN-γ production relative to IL-4—is critical for effective anti-tumor immunity. IFN-γ inhibits metastasis and angiogenesis while promoting cancer cell apoptosis [92]. Furthermore, IFN-γ amplifies the immune response by activating NK cells, which produce additional IFN-γ, driving dendritic cell maturation and Th1 differentiation, enhancing the cytotoxic activity of CD8^+^ T cells, and promoting M1 macrophage [92,93].

Furthermore, α-GalCer has been shown to induce anergy in iNKT after repeated administration in mice, accelerating metastasis [94]. On the other hand, clinical trials using α-GalCer generate suboptimal immune responses having limited efficacy [16,95]. Considering all this, the design of glycolipid agonists of iNKT cells that produce a strong Th1-biased response with increased IFN-γ production without affecting iNKT cell activation has been challenging. Several researchers have pursued this goal to synthesize structural analogs of α-GalCer having these properties [15,96]. Among these, the most studied include α-C-GalCer, 7DW8-5, AH10-7, C34, and non-glycosidic threitolceramide (ThrCer)-based analogs.

α-C-GalCer has a CH_2_-based glycosidic linkage rather than the oxygen-based glycosidic linkage of α-GalCer, which would prevent its degradation by glycosidases, increasing the stability of the compound [97]. This ligand promotes a strong serum production of IFN-γ and IL-12, with almost non-detectable production of IL-4 and TNF-α compared to α-GalCer [98]. Interestingly, this glycolipid promoted higher upregulation of CD40L on CD1d-dimer^+^ CD19^-^ iNKT cells. DCs loaded with α-C-GalCer protected against mice lung metastases of B16 melanoma, generating a more significant effect than a five-fold higher dose of α-GalCer [98]. Compared to α-GalCer, 7DW8-5 has a fluorinated benzene ring at the end of a C8-length fatty acyl chain [99]. This glycolipid promotes a higher activation of iNKT cells and IFN-γ production, eliciting anti-tumoral effects against mice medulloblastoma and human breast cancer cell lines in vitro and in vivo when transferred to immunodeficient mice [99,100].

As for AH10-7, this glycolipid is modified in the galactose, with a hydrocinamoyl ester group on carbon 6. It lacks the hydroxyl group on carbon 4 of the sphingosine, generating a Th1-biased production of cytokines, together with strong anti-tumor activity against B16-F10 melanoma (which is more aggressive and spreads to several tissues) compared to α-GalCer, which also generated this effect in humanized mice hCD1d-KI, which express human CD1d [101,102]. Like AH10-7, analogs with amide-linked phenyl alkane substitutions on the C4″-position of the galactose ring are also prone to Th1 polarization of iNKT cells, inducing higher T-bet but lower granzyme B compared to α-GalCer. Despite this, it causes strong anti-tumoral effects against B16-F10 melanoma [103].

C34 analogue contains two phenyl rings on the acyl chain compared to α-GalCer and elicited a strong IFN-γ production without inducing iNKT cell anergy. Additionally, its anti-tumoral activity has been addressed in mice breast, lung, melanoma, and neuroblastoma cancer [104,105]. Interestingly, it has been shown that α-GalCer generates an accumulation of immunosuppressive myeloid-derived suppressor cells (MDSCs) in the spleen, which might attenuate anti-tumor efficacy, whereas C34 does not create this effect [106,107].

Despite the advances regarding the effectiveness of α-GalCer analogs in the induction of anti-tumoral effects, not all of these have been evaluated in the context of human cancer. Implementing clinical trials is challenging, expensive, and limited to a cohort of subjects. Therefore, PDOs represent a novel and suitable approach to evaluate their immunotherapeutic capacity. Data show that the in vitro response of PDOs is predictive of patient response to therapy, therefore holding promising results for personalized treatment [27].

#### 3.2.2. Study of Dynamics and Interaction Between Tumoral and Immune Cells in the TME

Understanding TMEs is critical for the development of appropriate immunotherapy. This requires studying their dynamics and cellular composition, which includes malignant and non-malignant cells, such as fibroblasts, endothelial cells, and immune cells embedded in the ECM [54]. Another key concept is that the tumor is a collection of heterogeneous resident and infiltrating cells, which evolves continuously.

The TME comprises innate and adaptive immune cells, performing pro- or anti-tumorigenic functions depending on environmental signals (Figure 2). Tumor-infiltrating T cells are heterogeneous and have distinct specificities influencing tumorigenesis. TMEs fall into three categories: immune infiltrated tumor, which indicates an active immune response; immune excluded tumor, indicating that T cells are located in the periphery with no infiltration; and immune silent tumor, with no immune cell infiltrates [36]. CD8^+^ T cells are associated with a favorable prognosis in cancer patients, and in addition to killing tumor cells, IFN-γ production suppresses angiogenesis. CD4^+^ T cells with Th1 phenotype support CD8^+^ T cell function through IL-2 and IFN-γ production. However, acquiring a Treg phenotype promotes tumor development and progression and supports tumor immune escape [36]. Several Treg subsets have been reported: Tregs derived from thymic selection (tTregs), peripherally converted Tregs (pTregs), tissue-resident Tregs (tr-Tregs), and follicular Tregs (Tfr). However, their role in cancer immune response is unclear [9].

B cells also support T cell function through antigen-presentation or IFN-γ production. In other cases, B cells may acquire a regulatory phenotype secreting IL-10 and TGF-β, which ultimately promote tumor aggression and immunosuppression in macrophages, neutrophils, and cytotoxic T cells [108]. Innate immune cells also alternate between anti- and pro-tumoral roles, such as macrophages switching from M1 to M2 cells, neutrophils from ROS to MMP-9 and VEGF production, and DCs switching from TNF-α and IL-6 production to a tolerogenic phenotype [36].

In this context, iNKT cells are widely known for their immunomodulatory capacity and ability to transactivate immune cells. As mentioned previously, the induction of Th1-biased immune response is one of the main goals in iNKT cell-based immunotherapy, which could induce an anti-tumoral immune microenvironment, inducing immune cell infiltration, together with the activation of NK cells, Th1 cells, and cytotoxic CD8^+^ T cells [77].

Distinct tumor-infiltrating and tumor-resident cells express CD1d. However, the exact dynamics and interactions between these and iNKT cells are not fully understood. The implications of CD1d expression in cancer cells are controversial. Based on a recent revision, loss of CD1d expression promotes tumorigenesis in the context of chronic myeloid, leukemia, cervical cancer, lung cancer, gastrointestinal cancer, prostate cancer, and pancreatic cancer, whereas in breast cancer, thyroid cancer, glioblastoma, medulloblastoma, and renal cancer, gain of CD1d expression promotes tumorigenesis [14]. A versatile approach for evaluating this dynamic could involve using PDOs, given the various strategies available for developing organoids associated with different types of cancer [109].

On the other hand, we previously addressed the induction of anergy in iNKT cells when administrating α-GalCer into mice. A similar phenotype is observed in patients, where there is a reduction of circulating iNKT cells, which also have reduced proliferative capacity and phenotypes that differ from Th1. This phenotype has been proposed to result from chronic stimulation due to altered lipid presentation in tumorigenesis. These alterations include the accumulation of lipid bodies, which negatively affects the localization of MHC-I and CD1d in the plasma membrane; higher CD1d expression in DCs due to polyunsaturated fatty acids (PUFAs)-induced PPAR-γ; and alteration of membrane dynamics interfering with CD1d–lipid-antigen conjugation rate between APC and iNKT cells [110]. By using PDOs, it would be possible to evaluate iNKT cell activation during the natural course of the disease to access the stage at which the anti-tumoral capacity of iNKT decreases. Additionally, CD1d-mediated lipid presentation during cancer development would allow the identification of endogenous agonists and further strategies to reduce their presentation, improving iNKT cell activation through exogenous stimulation [111].

#### 3.2.3. Co-Culture of PDOs with Autologous Immune Cells

Co-culturing PDOs with autologous immune cells, such as PBMCs or TILs, replicates immune–tumor dynamics in physiologically relevant 3D models. These systems retain crucial aspects of the TME, including immune heterogeneity, cytokine secretion profiles, and mechanisms of immune evasion [47,50]. For instance, Ding et al. (2023) developed patient-derived micro-organospheres (MOS), a variation of PDOs, to preserve tumor-resident immune cells and assess immunotherapeutic responses. Their platform enabled the evaluation of T cell-mediated cytotoxicity and responses to immune checkpoint inhibitors (ICIs) such as nivolumab, demonstrating the ability of MOS to maintain functional immune interactions within the TME [47].

In this context, research by Jenkins et al. (2018) underscores the relevance of ex vivo platforms, such as PDOs, in preserving the immune contexture of tumors. The study demonstrates the capacity of PDOs to retain key immune cell populations, including CD8^+^ T cells, and their interactions with tumor cells, which are pivotal for evaluating immune checkpoint blockade therapies [112]. By integrating PDOs with immune cells in three-dimensional microfluidic systems, the study provides a dynamic environment to model responses to immunotherapies like PD-1 blockade. These platforms enable real-time profiling of cytokine responses and immune cell infiltration, which are critical for understanding therapeutic resistance and optimizing combination therapies.

Moreover, Carrese et al. (2024) explored the role of LAG3, an immune checkpoint molecule, in breast cancer PDOs. Their study demonstrated that blocking LAG3 using relatlimab restored T cell-mediated cytotoxicity, reduced immune suppression, and increased the production of pro-inflammatory cytokines like IFN-γ and IL-12. This underscores the potential of PDOs for testing immune-targeted therapies while retaining the immune and genetic heterogeneity of the tumor [113].

These co-culture systems have also proven effective in personalized immunotherapy development. For instance, Parikh et al. (2023) employed PDOs derived from colorectal and breast cancers to analyze TIL responses to neoantigens, enabling personalized T-cell reactivity studies [114]. Furthermore, Neal et al. (2018) utilized air–liquid interface (ALI) organoids to maintain diverse immune cell populations, including tumor-associated macrophages, NK cells, and T cells, which were instrumental in modeling responses to ICIs [6]. Although the synergy between this approach and iNKT cells has not been directly studied in PDOs, their integration could amplify anti-tumor immune responses, as suggested by their individual mechanisms of action.

These systems also provide opportunities to model the suppression mechanisms tumors employ to evade immune responses. For example, pancreatic ductal adenocarcinoma (PDAC) PDOs co-cultured with autologous immune cells have revealed immune suppression, including increased Treg activity and diminished effector T cell responses, aligning with clinical observations [73]. Additionally, PDO systems have demonstrated their utility in studying the interactions between various immune cells and the TME. These platforms facilitate the integration of immune components, such as CD4^+^ and CD8^+^ T cells, NK cells, and dendritic cells, within the organoid matrix, enabling real-time analysis of immune–tumor dynamics. For example, iterative co-cultures between PBMCs and PDOs have been shown to induce tumor-specific T cell responses, enhancing the understanding of immune cell reactivity against tumors [31,73].

While PDO platforms have been proposed as a tool for modeling iNKT cell interactions with the TME, further studies are needed to substantiate this application. This integration of immune cells and PDOs underscores their potential to predict patient responses, optimize therapeutic approaches, and advance our understanding of immune-oncology in diverse tumor settings.

#### 3.2.4. Combined Therapy Between iNKT Cells and Immune Checkpoint Inhibitors

The combination of iNKT cells and immune checkpoint inhibitors in the context of PDOs represents a cutting-edge strategy in cancer immunotherapy.

As mentioned previously, the TME often generates immunosuppression. Therefore, immune checkpoint inhibitors, such as PD-1/PD-L1 and CTLA-4 blockers, have revolutionized cancer immunotherapy, being approved for many types of cancer [23,66]. When integrated with iNKT cell therapies in PDO models, these inhibitors could amplify anti-tumor responses by overcoming tumor-mediated immune suppression. Combining these therapies could target both the adaptive and innate immune components, providing a synergistic approach to cancer treatment [33].

A critical advantage of this integrated approach is the ability to model and predict patient-specific responses to therapy in PDOs. For instance, using α-GalCer-loaded dendritic cells to stimulate iNKT cells and immune checkpoint inhibitors might reveal promising results in preclinical PDO models, demonstrating tumor suppression [27]. This area is rapidly evolving, with ongoing investigations focusing on enhancing the efficacy and safety of these therapies. Integrating PDO models in preclinical studies provides an invaluable tool to bridge the gap between laboratory findings and clinical application, offering a pathway toward more precise and effective cancer immunotherapies.

#### 3.2.5. CAR-NKT Cells in Immunotherapy

Chimeric Antigen Receptor Natural Killer T (CAR-NKT) cells represent an innovative platform in cancer immunotherapy. These cells combine the natural properties of NKT cells with the targeted cytotoxicity of CARs, providing a unique dual mechanism of action [115].

By engineering NKT cells to express CARs targeting specific tumor antigens, such as CD19 or GD2, researchers have enhanced their specificity and potency against cancers [116]. This innovation has significant advantages over other CAR-based therapies. For instance, CAR-NKT cells demonstrate a reduced risk of graft-versus-host disease (GvHD), making them suitable for allogeneic (off-the-shelf) therapies [117]. They also show superior trafficking to tumors and possess the ability to modulate immune responses within the TME. Additionally, they exhibit a lower incidence of severe side effects like cytokine release syndrome (CRS), commonly associated with CAR-T therapies [118].

Preclinical studies have demonstrated the efficacy of CAR-NKT cells in treating both hematologic malignancies, such as CD19^+^ B-cell lymphomas, and solid tumors, including neuroblastoma and GD2+ cancers [118]. Early clinical trials have reported promising outcomes, with notable tumor regression and minimal toxicities in patients [119]. Despite these advances, challenges remain, including optimizing CAR designs for better persistence, overcoming immunosuppressive signals within tumors, and scaling up production for widespread clinical use.

Future directions for CAR-NKT cell therapy include combining it with immune checkpoint inhibitors or cytokine support in a PDO model to enhance efficacy and per-sistence, similar to that which has been reported for CAR-T cells [120]. Researchers are also exploring universal CAR-NKT products to make this therapy more accessible. With their unique properties and growing evidence of effectiveness, CAR-NKT cells hold significant promise as a next-generation immunotherapy for hematologic and solid tumors.

## 4. Challenges and Innovations in PDO–iNKT Integration

The integration of iNKT cells with PDO platforms presents significant opportunities and challenges in advancing cancer immunotherapy. While the use of iNKT cells in PDO systems has not yet been extensively explored, this represents a fertile area for innovation. The potential for modeling iNKT cell interactions with the TME within PDOs could provide transformative insights into immune–tumor dynamics and therapeutic strategies.

### 4.1. Technical Challenges

One of the primary challenges lies in maintaining the genetic, epigenetic, and functional fidelity of PDOs during long-term culture. Prolonged cultivation can result in genetic drift, loss of tumor heterogeneity, and alterations in key signaling pathways, which compromise the relevance of the models. Advanced cryopreservation techniques and optimized culture conditions are being developed to address these issues and ensure the reproducibility of PDO-based studies. Additionally, incorporating iNKT cells into these platforms requires sophisticated co-culture systems that preserve their viability and functionality over time.

Replicating the complex TME within PDO systems is another critical challenge. The TME comprises diverse cellular and molecular components, including immune cells, stromal cells, and extracellular matrix elements, that influence tumor progression and immune responses. Successfully integrating iNKT cells into PDOs will require advanced 3D scaffolds, microfluidic technologies, and co-culture methodologies to mimic these intricate interactions. Moreover, the ability to maintain CD1d expression on tumor cells, which is crucial for iNKT cell recognition, will be essential for studying their therapeutic potential.

Standardization and scalability also present hurdles. Variability in PDO culture protocols across laboratories could limit the reproducibility and translational potential of research. Furthermore, scaling PDO–iNKT platforms for high-throughput drug screening will necessitate the development of automated systems and quality control measures. Emerging solutions, such as robotics-assisted organoid handling and machine learning algorithms, hold promise for addressing these challenges.

### 4.2. Innovative Approaches

To overcome these challenges, several innovative approaches could enhance the integration of iNKT cells with PDOs:

**Single-Cell Technologies**: Advanced single-cell RNA sequencing and spatial transcriptomics could provide detailed insights into the interactions between iNKT cells and PDOs. These tools can elucidate cytokine signaling pathways, immune cell infiltration dynamics, and tumor-intrinsic resistance mechanisms, offering a high-resolution view of tumor–immune interactions. By identifying phenotypic changes in iNKT cells, these techniques could inform the development of more effective therapeutic strategies.

**CRISPR-Cas9 Gene Editing**: CRISPR-based tools could enable precise genetic modifications in PDOs and iNKT cells to enhance their interactions. For example, upregulating CD1d expression on tumor cells could improve iNKT cell recognition and cytotoxicity. Additionally, genetic knockouts or modifications could help elucidate resistance mechanisms and identify novel therapeutic targets.

**Organoid-on-a-Chip Systems**: Microfluidic platforms that mimic the vascular and lymphatic networks of tumors could provide a dynamic environment for studying iNKT cell behavior, including migration, activation, and tumor cell lysis. These systems could also facilitate real-time monitoring of therapeutic responses under conditions that closely simulate in vivo scenarios.

**Immunomodulatory Agents**: Incorporating cytokines such as IL-12, IL-15, and IL-21 into PDO–iNKT co-cultures could enhance iNKT cell activation and persistence. Combining these cytokines with immune checkpoint inhibitors may amplify anti-tumor immune responses. Emerging methodologies for sequential or concurrent application of these agents could optimize treatment regimens and provide valuable data for clinical translation.

**Bioprinting and Custom Scaffolds**: Advances in bioprinting technology could enable the creation of tailored scaffolds that replicate the extracellular matrix of specific tumor types. These scaffolds could support the integration of iNKT cells, enhancing the physiological relevance of the models and providing a more accurate platform for studying immune–tumor interactions.

### 4.3. Addressing Immunosuppression in the TME

The immunosuppressive nature of the TME poses a significant barrier to the effectiveness of iNKT cell therapies. Tregs, MDSCs, and tumor-associated macrophages (TAMs) collectively create an environment that inhibits iNKT cell function through mechanisms such as cytokine secretion (e.g., TGF-β, IL-10), metabolic competition, and immune checkpoint signaling. This multifaceted suppression reduces the efficacy of immunotherapies targeting tumor-associated immune cells [48,121].

PDO models offer a unique platform for studying these interactions and testing novel strategies to counteract immunosuppression. For example, targeting TAMs with CSF-1R inhibitors has demonstrated potential in reducing pro-tumoral polarization of macrophages, thereby restoring effective anti-tumor immune responses [121,122]. Similarly, blocking MDSC recruitment through CCR2 antagonists has shown promise in preclinical studies for enhancing immune cell infiltration and function [123].

The modulation of the metabolic environment within the TME is another promising avenue. Reducing lactic acid accumulation, a byproduct of anaerobic glycolysis often associated with immune suppression, or targeting glutamine metabolism, crucial for MDSC and TAM survival, can enhance the activity and persistence of iNKT cells in co-culture systems. PDO-based studies can help refine these approaches by providing dynamic and patient-specific insights [121].

Furthermore, combining metabolic modulation with cytokine-based therapies, such as the administration of IL-12 and IL-15, can synergistically boost iNKT cell activation and proliferation. These cytokines not only enhance cytotoxic activity but also promote the recruitment of other immune effector cells, such as CD8^+^ T cells and NK cells, into the TME. Advanced PDO models co-cultured with iNKT cells provide a robust framework for evaluating these strategies in real time, ensuring translational relevance [13,48].

By enabling a comprehensive analysis of tumor–immune interactions, PDO systems pave the way for the development of combination therapies designed to overcome immunosuppression. These platforms support the optimization of treatment regimens that integrate metabolic inhibitors, checkpoint blockade, and cytokine therapies, ultimately enhancing the clinical efficacy of iNKT cell-based immunotherapies [28].

### 4.4. Future Directions

One significant focus should be on developing automated PDO–iNKT systems capable of high-throughput screening of extensive drug libraries and therapeutic combinations. These systems could accelerate the discovery of novel cancer treatments by reducing variability and improving scalability through robotics and microfluidic technologies [13,28]. In parallel, dynamic imaging techniques such as intravital microscopy and three-dimensional fluorescence imaging should be incorporated to enable real-time visualization of iNKT cell interactions with PDOs. These tools provide critical insights into immune cell infiltration, activation, cytokine signaling, and exhaustion dynamics within the TME [62].

Expanding the diversity of PDO models is equally crucial for broadening the scope of research. Current PDO repositories predominantly focus on common cancer types, leaving rare cancers and metastatic lesions underrepresented. By developing PDOs derived from these understudied tumor types, researchers can explore unique tumor–immune interactions and identify novel therapeutic targets [12,13]. Additionally, creating PDOs from treatment-resistant and recurrent tumors will facilitate the study of resistance mechanisms and enable the design of adaptive therapeutic strategies [14,62].

The integration of multi-omics approaches, including genomics, transcriptomics, proteomics, and metabolomics, offers promising opportunities for advancing PDO–iNKT systems. These technologies can uncover biomarkers of response or resistance, elucidate mechanisms of iNKT cell activation and exhaustion, and guide the development of precision medicine strategies. Furthermore, the use of artificial intelligence (AI) and machine learning algorithms to analyze these datasets could enhance the prediction of therapeutic outcomes and support patient stratification for personalized treatments [28].

Addressing immunosuppression in the TME remains a critical research priority. Effective strategies, including metabolic reprogramming to improve iNKT cell fitness, modulation of TAMs and MDSCs, and the use of cytokines such as IL-12 and IL-15 to boost iNKT cell activity, have shown promise. For example, PDO models provide an innovative platform for evaluating these approaches under physiologically relevant conditions, enabling optimization of therapies to counteract immune suppression and enhance therapeutic efficacy [14,48].

Collaborative efforts between academia, industry, and clinical institutions are essential to bridge the gap between preclinical research and clinical application. Standardizing protocols for PDO generation, iNKT cell integration, and therapeutic testing will enhance reproducibility and reliability across studies. Partnerships with biopharmaceutical companies could accelerate the translation of PDO–iNKT research into clinical trials, expediting the development of innovative therapies for cancer patients [121,124].

In conclusion, the future of PDO–iNKT research holds immense potential for transforming cancer immunotherapy. By addressing these challenges and embracing technological innovations, PDO–iNKT platforms can significantly enhance our understanding of tumor biology, improve treatment outcomes, and provide immunotherapeutic approaches.

## 5. Conclusions

In this article, we have highlighted the transformative potential of integrating iNKT cells with PDO platforms to advance cancer research and immunotherapy. iNKT cells are a unique subset of T cells capable of bridging innate and adaptive immunity. They exert potent anti-tumor effects by secreting cytokines such as IFN-γ, IL-4, and TNF-α, engaging in direct cytotoxicity, and recruiting immune effectors like CD8^+^ T cells, DCs, and NK cells. Despite their promise, the efficacy of iNKT cell-based therapies is often compromised by the immunosuppressive tumor microenvironment (TME), which includes Tregs, myeloid-derived suppressor cells (MDSCs), and tumor-associated macrophages (TAMs). This review underscores the utility of PDO platforms in overcoming these barriers and enhancing our understanding of iNKT cell interactions with the TME.

PDOs offer a physiologically relevant model that preserves the architecture, heterogeneity, and genetic integrity of primary tumors. When integrated with iNKT cells, PDOs provide an unparalleled platform for studying tumor–immune dynamics in real time. These systems enable the testing of Th1-biased immune responses driven by α-GalCer analogs, which have been shown to enhance iNKT cell-mediated anti-tumor activity. Furthermore, PDOs allow for the exploration of metabolic modulation strategies, such as targeting glutamine metabolism and reducing lactic acid accumulation, to improve iNKT cell fitness and persistence. Studies leveraging these platforms can elucidate the effects of cytokine adjuvants like IL-12 and IL-15 in enhancing iNKT cell activation and the synergistic potential of combining iNKT cells with immune checkpoint inhibitors.

Technological innovations such as CRISPR-Cas9 gene editing, organoid-on-a-chip systems, and multi-omics approaches significantly enhance the functionality and utility of PDO–iNKT platforms. These tools enable precise genetic modifications, dynamic modeling of immune cell behavior, and biomarker identification, facilitating personalized and high-throughput immunotherapy research. By integrating advanced imaging modalities like single-cell transcriptomics and intravital microscopy, researchers can gain critical insights into the spatial and temporal dynamics of iNKT cell interactions within the TME.

While the integration of iNKT cells with PDOs is still in its early stages, the potential applications are vast. PDO–iNKT systems are particularly valuable for studying rare and treatment-resistant cancers, broadening the scope of personalized medicine. These platforms also hold promise for optimizing therapies that target immunosuppressive mechanisms, such as TAM polarization and MDSC recruitment, and for developing next-generation immunotherapies like CAR-iNKT cells.

Despite the significant progress achieved, challenges remain. Standardizing protocols for PDO generation, iNKT cell integration, and therapeutic testing is critical to ensure reproducibility and scalability. Expanding PDO repositories to include diverse tumor types, including rare and metastatic cancers, is equally important. Collaborative efforts between academia, industry, and clinical institutions will be essential to bridge the gap between preclinical research and clinical applications, accelerating the translation of PDO–iNKT platforms into clinical trials.

In conclusion, the integration of PDOs and iNKT cells represents a powerful framework for addressing the complexities of tumor biology and developing effective cancer immunotherapies. By embracing technological advancements and fostering interdisciplinary collaborations, PDO–iNKT platforms have the potential to revolutionize cancer treatment, offering hope for improved patient outcomes and personalized therapeutic strategies.

## Figures and Tables

**Figure 1 cancers-17-00406-f001:**
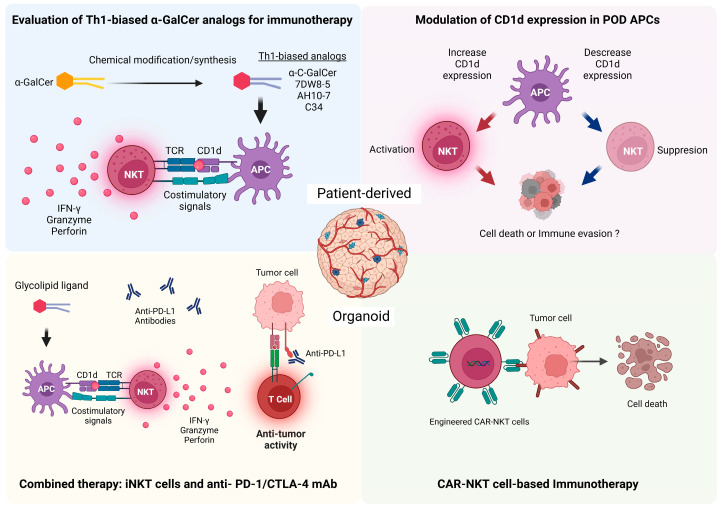
Possible applications of iNKT cells in PDOs for cancer immunotherapy assessment. Patient-derived organoids provide a suitable platform for evaluating immunotherapeutic approaches, with several advantages compared to 2D culture assays and mice models. Unlike these, organoids maintain the physiological characteristics of the tumor microenvironment, with the ability to closely replicate patient-specific tumor biology and, therefore, personalized treatments. NKT cells combine innate and adaptive properties and can respond against glycolipid antigens presented by APCs in the CD1d molecule, triggering the secretion of diverse cytokines and cytotoxic molecules. Therefore, these cells not only have a direct anti-tumoral effect but also modulate the function of immune cells infiltrated in the TME, orchestrating the anti-tumoral response. Some applications of iNKT cells on PDOs include the evaluation of novel Th1-biased α-GalCer analogs for immunotherapy; modulation of CD1d expression in APCs and tumor cells and assessment of their effect in tumor cells; combined NKT cell therapy with immune checkpoint inhibitors such as anti-PD-1/PD-L1 and anti-CTLA monoclonal antibodies (mAb); Use of CAR-NKT for the elimination of tumor cells.

**Figure 2 cancers-17-00406-f002:**
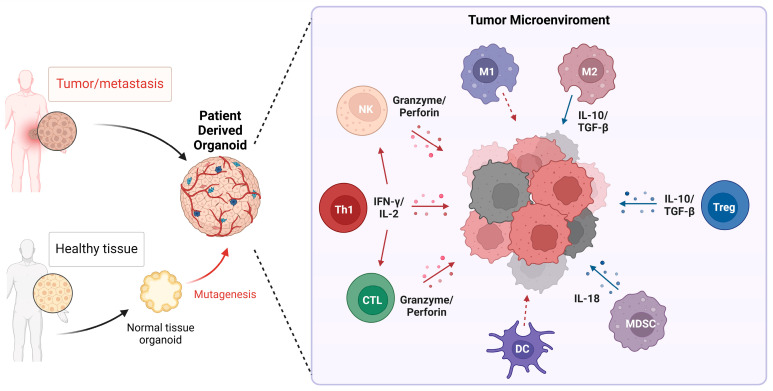
A patient-derived organoid replicates the tissue’s architecture, function, and cellular heterogeneity of origin, including the tumor microenvironment. Patient-derived organoids (PDO) can be generated from tumoral or healthy tissues, and in this second case, they can be turned cancerous through mutagenesis. Incorporation of autologous immune cells into a PDO facilitates the study of cellular dynamics and interaction with tumoral cells, providing a close replicate of the patient-specific tumor biology, allowing prediction of the possible outcome, being, therefore, an effective method for developing cancer immunotherapies. Infiltrating immune cells are widely heterogeneous, which might include cells with anti-tumoral or pro-tumoral roles. T helper 1 (Th1) cells are responsible for the production of IFN-γ and IL-2, which, despite suppressing angiogenesis and inducing tumoral cell death, also promotes the activation of cytotoxic CD8^+^ T cells (CTL) and Natural Killer (NK) cells, both of which secrete cytotoxic enzymes such as granzyme B and perforin, triggering tumoral cell death. In addition, IFN-γ stimulates macrophage differentiation into the M1 phenotype with anti-tumoral activity. In line with this, there is cell uptake of DCs, which present tumoral antigens for further presentation to T cells, sustaining this complex dynamic. However, some cells promote tumoral development when they acquire an immunosuppressive phenotype, such as regulatory T cells (Tregs) or M2 macrophages, producing IL-10 and TGF-β. Similarly, myeloid-derived suppressor cells (MDSCs) are widely described in the tumoral context, which produces IL-18, favoring its accumulation in the tumor, and mediates suppression of CD4^+^ and CD8^+^ T cells, favoring tumor development.

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
