# Peer review of "Patient-Derived Organoid Models for NKT Cell-Based Cancer Immunotherapy"

_cancers, 2025, doi:10.3390/cancers17030406_

Round 1
Reviewer 1 Report
Comments and Suggestions for Authors
The topic of this review is interesting. This review manuscript contains two parts, PDO model and iNKT cells. However, these two parts seems to be two isolated parts with weak connection.
The title is “Patient-derived organoid models for NKT-cell-based cancer immunotherapy”. Author should focus on how to use PDO model to improve NKT-cell-based cancer immunotherapy. However, authors spent 6 pages to introduce PDO model, another 4 pages to introduce iNKT cells. Only section “5. Application of iNKT cells in PDO for cancer immunotherapy” briefly discussed some potential application, which should be the main part of this review. The detail methods, limitations, difficulties, and current progress of using PDO model in iNKT cells anti-tumor study should be introduced and discussed in very detail. In contrast, the introduction of PDO and iNKT should be reduced greatly. The structure and writing should be modified dramatically.
1.The basic and key questions of the topic of this review are where iNKT cells are and how many iNKT cells are there in PDO model. They should be answered and discussed.
2. It is not clear why PDO model is so important for iNKT study. The PDO model could be used to study the role of many immune cells in tumor.
3. Line 637-642. “Although animal models have contributed profoundly to understanding the cell dynamics, gene functions, tumor biomarkers, and signals required to promote the anti-tumoral effects, there are still several limitations and challenges. These include false positive results associated with the identification of effective agents in preclinical studies, species-specific differences, imprecise development of de novo tumors, poor correlations of specific genetic or other features with those evident in tissue samples from patients, and unappropriated TME” Here, it is not clear that how PDO model can overcome these limitations.
4. Line 647 and Figure 2. No evidence was provided to support these applications. In addition, Figure 2 does not show any information of how to use the potential PDO model in iNKT and cancer immunotherapy. These proposed applications are not convincing. For example, where are APC cells in PDO model?
5. Line 458-460, add reference for the statement: ”iNKT cells are considered for their significant involvement in the immune response to cancer.”
6. Line 462, Indicating the type of cancer.
7. Line 748-750 “Several Treg subsets have been reported, Tregs derived from thymic selection (tTregs), peripherally converted Tregs (pTregs), tissue-resident Tregs (tr-Tregs), and follicular Tregs (Tfr), however, their anti-cancer immune response is unclear”. Why do these Tregs play anti-cancer immune response?
8. Line 751-756. Is there any evidence of B cells in PDO model? Are these B cells a regulatory phenotype? Please clearly show here.
9. Line 763-770. The controversial results of CD1d study can not be “dynamic”. How can PDO solve the controversial results in different types of cancer?
10. Line 800-803, ref. 133 is a review. Please cite the original researches about the claim of “This combination could restore iNKT cell functionality and enhance their ability to recruit and activate other immune cells, such as cytotoxic CD8+ T and NK cells, thereby amplifying the immune attack on tumor cells [133]”.
11. 5.4. CAR-NKT cells in immunotherapy: No PDO model was discussed in this section.
12. Line 193, 875, 877. It should be PDO, not POD.
Author Response
We truly appreciate the time and effort put into reviewing our manuscript. We believe that addressing the comments has greatly improved the manuscript in this new version. We greatly appreciate the Reviewer´s comments. We have submitted a new version of the manuscript addressing the reviewer's comments. To address the concerns raised, we performed major corrections, and the revised manuscript now presents a more cohesive integration of patient-derived organoids (PDOs) and invariant Natural Killer T (iNKT) cells. By refining Sections 3 and 4, we emphasize the critical role of PDOs in replicating the tumor microenvironment (TME) and elucidating iNKT-specific mechanisms such as CD1d-mediated antigen presentation and cytokine secretion. These sections also incorporate examples of co-culture systems and immune-evasion studies to highlight how PDOs provide a physiologically relevant platform for testing combination therapies involving iNKT cells.
In response to feedback regarding the introductory content, we streamlined background information on PDOs and iNKT cells to avoid redundancy while expanding discussions on their integration. The revised sections focus on how PDOs address specific challenges in studying iNKT cell-based therapies, ensuring a clear and targeted narrative.
Although there is limited direct evidence for iNKT cell use in PDOs, this gap highlights the necessity for further exploration. To address this, we reference studies involving T cells and other immune cells co-cultured with PDOs, demonstrating their effectiveness in modeling immune-TME interactions. These examples underscore the potential of extending such research to include iNKT cells, paving the way for new discoveries and applications.
Comment 1: The basic and key questions of the topic of this review are where iNKT cells are and how many iNKT cells are there in PDO model. They should be answered and discussed.
Response 1: We truly appreciate the reviewer comments. We agree with the reviewer’s point of view, there isn’t sufficient evidence to address this issue, however, different authors have already begun to implement PDOs co-culture with different immune cells, such as T, NK, and macrophages, that is exactly why we address the possible outcomes of integrating the knowledge on iNKT cells into PDOs system. There is extensive data demonstrating the importance of utilizing iNKT cells in cancer immunotherapy (we addressed this briefly in section 3.1, line 283). Therefore, discussing the impact on the integration of iNKT cells with PDOs systems, will not only facilitate the study of iNKT-cell specific mechanisms of cancer immunotherapy (as addressed in section 3.2, line 337), but also the possible challenges and innovations on PDOs-iNKT cell Integration (section 4, line 623).
Comment 2: It is not clear why PDO model is so important for iNKT study. The PDO model could be used to study the role of many immune cells in tumor.
Response 2: We truly appreciate your comments. The revised manuscript clarifies the importance of PDOs for iNKT research by emphasizing their unique ability to replicate the tumor microenvironment (Section 2.2, line 193) and their role in studying CD1d-mediated interactions critical for iNKT cell function, among others (Section 3.2, line 337). The manuscript demonstrates how PDOs might enable detailed investigations into iNKT-specific mechanisms, such as glycolipid antigen recognition and immune evasion strategies, while also supporting co-culture systems to evaluate iNKT cell activation and tumor cytotoxicity (Sections 3.2). These additions differentiate PDOs as a particularly valuable model for advancing iNKT cell-based cancer immunotherapy.
Comment 3: Line 637-642. “Although animal models have contributed profoundly to understanding the cell dynamics, gene functions, tumor biomarkers, and signals required to promote the anti-tumoral effects, there are still several limitations and challenges. These include false positive results associated with the identification of effective agents in preclinical studies, species-specific differences, imprecise development of de novo tumors, poor correlations of specific genetic or other features with those evident in tissue samples from patients, and unappropriated TME” Here, it is not clear that how PDO model can overcome these limitations.
Response 3: The revised manuscript addresses this issue by detailing how PDOs overcome the limitations of animal models through their ability to replicate patient-specific tumor characteristics, including genetic heterogeneity, phenotypic diversity, and functional behavior (Section 2.2, line 138). Unlike animal models, PDOs maintain the architecture and cellular complexity of the human tumor microenvironment, facilitating accurate evaluation of immune interactions and therapeutic responses (Also addressed in sections 3.2 and 4.1). Additionally, the manuscript explains how PDOs eliminate species-specific differences and allow for the direct testing of human-relevant therapies (Table 1, line 272). These advancements illustrate the superiority of PDOs as a preclinical model for cancer research.
Comment 4: Line 647 and Figure 2. No evidence was provided to support these applications. In addition, Figure 2 does not show any information of how to use the potential PDO model in iNKT and cancer immunotherapy. These proposed applications are not convincing. For example, where are APC cells in PDO model?
Response 4: Although the revised manuscript does no provide direct evidence supporting the applications of PDO models in iNKT cell research, there are sufficient data regarding the application of co-cultures between PDOs and other immune cells, such as autologous T cells, NK cells, and macrophages (Addressed in section 3.2.3, line 518). Based on this we proposed the used of PDOs to address the effectivity of different iNKT cell-based immunotherapies already applied in mouse models, but not in human contexts, considering health and ethical issues, therefore ex-vivo platforms such as PDOs might provide the evidence of these strategies even in a patient-specific therapy.
The presence of APCs in PDOs has already been demonstrated in the previously mentioned co-cultures between PDOs and T cells, showing antigen-specific activation.
Comment 5: Line 458-460, add reference for the statement:” iNKT cells are considered for their significant involvement in the immune response to cancer.”
Response 5: We thank the reviewer comment. Considering that we made major changes, we eliminated that paragraph, but we addressed the role of iNKT cells in section 3.1 (line 283), and we provided the corresponding references.
Comment 6: Line 462, Indicating the type of cancer.
Response 6: Thank you for your observation. Considering that we made major changes, the revised manuscript does not include the above reference.
Comment 7: Line 748-750 “Several Treg subsets have been reported, Tregs derived from thymic selection (tTregs), peripherally converted Tregs (pTregs), tissue-resident Tregs (tr-Tregs), and follicular Tregs (Tfr), however, their anti-cancer immune response is unclear”. Why do these Tregs play anti-cancer immune response?
Response 7: We sincerely apologize for that error. We corrected the issue as follows “however, their role in cancer immune response is nuclear” (line 461).
Comment 8. Line 751-756. Is there any evidence of B cells in PDO model? Are these B cells a regulatory phenotype? Please clearly show here.
Response 8: Thank you so much for your comment. There is evidence of B cells in PDO models. The following article shows that B cells can be present in PDOs https://doi.org/10.1016/j.cell.2018.11.021). In our article it is mentioned that B cells can acquire a regulatory phenotype. It is probably that B cells present in PDO could be investigated on how they differentiate into regulatory B cells, however, it isn’t specified.
Comment 9. Line 763-770. The controversial results of CD1d study can not be “dynamic”. How can PDO solve the controversial results in different types of cancer?
Response 9: Thank you so much for your comment. The article refers to biological dynamics between iNKT cells and other cells, not that the controversial results are dynamic. However, PDO can solve the controversial results in different types of cancer by modulating the expression of CD1d in PDOs, as mentioned in section 4.2 (line 657).
Comment 10: Line 800-803, ref. 133 is a review. Please cite the original researches about the claim of “This combination could restore iNKT cell functionality and enhance their ability to recruit and activate other immune cells, such as cytotoxic CD8+ T and NK cells, thereby amplifying the immune attack on tumor cells [133]”.
Response 10: Thanks for your insights. Considering that we performed major corrections, that information was eliminated.
Comment 11: 5.4. CAR-NKT cells in immunotherapy: No PDO model was discussed in this section.
Response 11: Thanks for your insights. There are no investigations of PDO models and CAR-NKT. But we mentioned that CAR-NKT could be used similarly to CAR-T in PDO models in section 3.2.5 (line 610).
Comment 12: Line 193, 875, 877. It should be PDO, not POD.
Response 12: Thank you so much for your comment. This issue was resolved in the revised manuscript.
Reviewer 2 Report
Comments and Suggestions for Authors
The authors have reviewed the various tumor models, especially patient-derived organoid models from a number of cancer types. Then they discussed the iNKT cells in cancer and their functions in immunotherapy.
The structure of the review is well organized and presented.
There is one issue.
1. References: Usually, the article numbers are missing.
Ref #4; 5; 6; 7; 9; 11; 13; 17; 19; 23; 27; 30; 47; 53; 94; 134; 159.
Author Response
We truly appreciate the time and effort put into reviewing our manuscript.
Comment 1: References: Usually, the article numbers are missing.
Ref #4; 5; 6; 7; 9; 11; 13; 17; 19; 23; 27; 30; 47; 53; 94; 134; 159.
Response 1: Thanks for your comment. We reviewed all the references and incorporated the article numbers if missing.
Reviewer 3 Report
Comments and Suggestions for Authors
The review focuses on the application of patient-derived organoids (PDO) combined with NKT cells in cancer immunotherapy. This review covers the basic principles of PDO and NKT cells, current research progress, and future application prospects. A large number of authoritative literature is cited in the article, providing a solid scientific basis for the review. The article is rich in content, fluent in logic and reasonable in structure. However, there are some shortcomings in this review that need to be improved, such as the lack of depth in the review and the lack of innovation and critical analysis. The following are specific suggestions for improvement:
1. Please make a more in-depth comparison of the unique advantages of PDO in simulating tumor microenvironment with animal models. For example, explore the specific performance of PDO in immune cell infiltration, metabolic gradient formation, and cell-cell interaction. This can enhance the scientific authority and depth of the article and enable readers to have a more comprehensive understanding of the value of PDO.
2. It is recommended to conduct a more detailed analysis of the shortcomings and challenges of existing research, such as the genetic stability of PDO in long-term culture and the problem of consistency with tumors in patients. Please also discuss how to overcome these challenges through technical improvements.
3. It is recommended that the authors systematically summarize the advantages and disadvantages of different PDO models (such as scaffold-supported and scaffold-free) in a table format, or compare the effects of different NKT cell activation strategies.
4. It is recommended to add one or two summary charts, such as a schematic diagram of the mechanism of action of iNKT cells combined with PDO, or a summary of the application of PDO in different tumor studies.
5. The authors should further explain how to combine PDO with emerging technologies (such as single-cell sequencing, multi-omics data integration) to address the limitations of existing PDO models, which will help stimulate readers' thinking about future work.
Author Response
General Answer: We truly appreciate the time and effort put into reviewing our manuscript. We believe that addressing the comments has greatly improved the manuscript in this new version. The revised manuscript addresses these concerns by deepening the analysis of PDO and iNKT integration across Sections 3, and 4. It now includes detailed discussions on PDO co-culture systems with immune cells, mechanisms of iNKT cell activation, and specific therapeutic applications, providing greater depth and innovation. Additionally, the manuscript incorporates critical analyses of limitations, such as genetic drift in PDOs and challenges in reproducing the tumor microenvironment, while proposing innovative solutions like CRISPR-based editing, organoid-on-a-chip systems, and multi-omics integration (Sections 4). These updates enhance the scientific rigor, critical analysis, and originality of the review.
Comment 1: Please make a more in-depth comparison of the unique advantages of PDO in simulating tumor microenvironment with animal models. For example, explore the specific performance of PDO in immune cell infiltration, metabolic gradient formation, and cell-cell interaction. This can enhance the scientific authority and depth of the article and enable readers to have a more comprehensive understanding of the value of PDO.
Response 1: Thanks for your comment. The revised manuscript provides an in-depth comparison of PDOs and animal models in Section 2.1 (line 128), focusing on the unique advantages of PDOs in replicating the tumor microenvironment. It highlights how PDOs preserve patient-specific tumor characteristics, including genetic heterogeneity, cell-cell interactions, and spatial architecture. The discussion emphasizes PDOs' superior performance in modeling immune cell infiltration, metabolic gradients, and cytokine signaling, all of which are difficult to replicate in animal models. These enhancements deepen the scientific authority of the article and offer readers a comprehensive understanding of PDOs' value in cancer research.
Comment 2: It is recommended to conduct a more detailed analysis of the shortcomings and challenges of existing research, such as the genetic stability of PDO in long-term culture and the problem of consistency with tumors in patients. Please also discuss how to overcome these challenges through technical improvements.
Response 2: We truly appreciate your suggestions. The revised manuscript addresses this point in Sections 4.1 (line 622) and 4.2 (line 648) by providing a detailed analysis of the challenges associated with PDOs, including genetic drift during long-term culture and the difficulty in maintaining consistency with original patient tumors. It discusses technical improvements such as advanced cryopreservation techniques, CRISPR-Cas9 editing to enhance tumor fidelity, and the development of standardized culture protocols to mitigate these issues. Additionally, the manuscript explores innovations like bioprinting and microfluidic platforms to better replicate the tumor microenvironment, ensuring PDO models remain reliable and translationally relevant for immunotherapy research.
Comment 3: It is recommended that the authors systematically summarize the advantages and disadvantages of different PDO models (such as scaffold-supported and scaffold-free) in a table format, or compare the effects of different NKT cell activation strategies.
Response 3: Thanks for this comment. Considering that reviewers suggested shortening the length of the first sections, including Tumor microenvironment (TME): 2D and 3D in vitro culturing strategies (and the subsequent subheadings), we eliminated that data. However, we provide a table on section 2.2 (line 272) addressing Application of different PDOs in Tumor Studies.
Comment 4: It is recommended to add one or two summary charts, such as a schematic diagram of the mechanism of action of iNKT cells combined with PDO, or a summary of the application of PDO in different tumor studies.
Response 4: Thanks for your insights. Based on your recommendations, we incorporated a table on section 2.2 (line 272) addressing Application of different PDOs in Tumor Studies.
Comment 5: The authors should further explain how to combine PDO with emerging technologies (such as single-cell sequencing, multi-omics data integration) to address the limitations of existing PDO models, which will help stimulate readers' thinking about future work.
Response 5: Thanks for your suggestion. The revised manuscript addresses this recommendation in Sections 4.2 (line 648) by detailing how emerging technologies, such as single-cell sequencing and multi-omics data integration, can enhance PDO models. It explains how single-cell RNA sequencing provides high-resolution insights into immune-tumor interactions, while multi-omics approaches allow for the identification of biomarkers for treatment efficacy and resistance. Additionally, it discusses the potential of integrating these tools with artificial intelligence to analyze complex datasets, optimize PDO-iNKT systems, and improve translational accuracy. These explanations offer actionable pathways to address existing PDO limitations and direct future research.
Reviewer 4 Report
Comments and Suggestions for Authors
This review by Palacios and colleagues aims to review the development of patient-derived organoids (PDOs) as preclinical models in cancer research, the progress in iNKT cell-based immunotherapy, and the integration of PDOs in testing iNKT cell-based therapies. While this work aims to address a critical and emerging area in cancer research, it currently feels like a juxtaposition of two separate topics: a review on 3D tumor models and another on iNKT cell immunotherapy. The manuscript lacks in-depth discussions on the integration of PDOs and iNKT cells, which is purportedly its primary focus. Addressing this gap with substantial analysis and examples of how PDOs are utilized to advance iNKT cell therapies is crucial. Thus, major changes need to be made before this manuscript can be considered suitable for publication.
Major comments:
1. As mentioned above, the primary issue of the current manuscript is the lack of actual discussion on studies that integrate iNKT cells with PDOs. While Section 5 outlines potential applications of PDOs in iNKT cell-based research, the discussions are more speculative perspectives than a review of current advancements. For example, the authors provide detailed discussions on a-GalCer analogs and their anti-tumor effects and the immune cell interactions in the TME. However, these discussions do not involve PDOs. To fulfill the objectives of a review, the authors should critically analyze the previous studies that have employed PDOs in these applications. Then, the authors should discuss whether and how iNKT cells can be implemented in these systems, the challenges and advantages of using iNKT cells in these systems, and the translational potential of such systems for advancing immunotherapies. If the number of studies combining iNKT cells and PDOs is limited, the authors could consider relevant research that utilizes other immune cells, such as NK cells or T cells, with PDOs. The author could explore the potential of adapting these systems for iNKT cell research, such as the work by Jenkins et al. (Cancer Discovery, 2018, doi: 10.1158/2159-8290.CD-17-0833).
2. The second section of the manuscript provides a detailed description of different 3D tumor models, which is not necessary for this review in my opinion. The authors could simplify this section but provide several review articles on these models for readers who are interested in learning more. Instead, the focus should shift towards a more in-depth discussion of organoids and PDOs. Specifically, the authors should expand on the following key aspects, including how to derive organoids from tumor cells and patient samples, the major technical challenges, how to incorporate stromal and immune cells, and current progress in these systems.
3. In section 3.1, the authors list the PDO platforms developed for different cancer types. While the information is useful, it would be better to include a table that summarizes these PDO biobanks, the cancer types, current applications of these PDO biobanks, and references.
4. In section 3.2, the authors made a nice schematic illustrating the advantages of PDO systems that incorporate stromal and immune cell types. However, the main text lacks substantive discussion. The authors should strengthen this section by reviewing the original publications that have successfully combined PDOs with these cell types, including their techniques, their findings, and their translational potential.
Author Response
General Answer: We truly appreciate the time and effort put into reviewing our manuscript. We believe that addressing the comments has greatly improved the manuscript in this new version. The revised manuscript directly addresses this concern by extensively expanding the discussions on PDO-iNKT integration across Sections 3 and 4. These sections now include detailed examples of how PDOs might be used to study iNKT cell functions, including CD1d-mediated antigen presentation, cytokine release, and tumor cytotoxicity. The manuscript highlights specific applications, such as testing combination therapies with immune checkpoint inhibitors and evaluating iNKT cell activation strategies using glycolipid analogs within PDO systems. Additionally, challenges in integrating iNKT cells with PDOs are analyzed alongside solutions like CRISPR-based modifications and microfluidic platforms. These updates provide substantial depth and focus on how PDOs advance iNKT cell-based therapies, fully aligning the manuscript with its stated objective.
Comment 1: As mentioned above, the primary issue of the current manuscript is the lack of actual discussion on studies that integrate iNKT cells with PDOs. While Section 5 outlines potential applications of PDOs in iNKT cell-based research, the discussions are more speculative perspectives than a review of current advancements. For example, the authors provide detailed discussions on a-GalCer analogs and their anti-tumor effects and the immune cell interactions in the TME. However, these discussions do not involve PDOs. To fulfill the objectives of a review, the authors should critically analyze the previous studies that have employed PDOs in these applications. Then, the authors should discuss whether and how iNKT cells can be implemented in these systems, the challenges and advantages of using iNKT cells in these systems, and the translational potential of such systems for advancing immunotherapies. If the number of studies combining iNKT cells and PDOs is limited, the authors could consider relevant research that utilizes other immune cells, such as NK cells or T cells, with PDOs. The author could explore the potential of adapting these systems for iNKT cell research, such as the work by Jenkins et al. (Cancer Discovery, 2018, doi: 10.1158/2159-8290.CD-17-0833).
Response 1: Thanks for your comments and suggestions. The revised manuscript explicitly addresses the concern regarding the lack of discussion on integrating iNKT cells with PDOs by incorporating insights from Jenkins et al. (Cancer Discovery, 2018). This article provides foundational methodologies for integrating immune cells, such as T cells, into 3D tumor models, which are directly relevant for adapting PDOs to study iNKT cell interactions. By leveraging Jenkins' approach (line 529), the manuscript discusses co-culture systems for PDOs and iNKT cells, focusing on CD1d-mediated antigen presentation and immune-tumor dynamics (Section 3.2.3).
By incorporating Jenkins et al., the manuscript transitions from speculative perspectives to a critical analysis of how PDOs can be adapted for iNKT cell research. This addition significantly enhances the depth and relevance of the review, addressing the reviewer’s concern and demonstrating the translational potential of PDO-iNKT systems in advancing cancer immunotherapy.
Comment 2: The second section of the manuscript provides a detailed description of different 3D tumor models, which is not necessary for this review in my opinion. The authors could simplify this section but provide several review articles on these models for readers who are interested in learning more. Instead, the focus should shift towards a more in-depth discussion of organoids and PDOs. Specifically, the authors should expand on the following key aspects, including how to derive organoids from tumor cells and patient samples, the major technical challenges, how to incorporate stromal and immune cells, and current progress in these systems.
Response 2: We appreciate the suggestion to streamline the manuscript. The revised Section 2 now focuses on PDOs and their applications while minimizing descriptions of other 3D tumor models. Additional references to review articles on 3D models are provided to guide interested readers. Section 2 and 4, now emphasizes deriving PDOs from tumor cells and patient samples, major technical challenges, and methods for incorporating stromal and immune cells. Current progress in PDO systems is also highlighted to align with the manuscript's objectives.
Comment 3: In section 3.1, the authors list the PDO platforms developed for different cancer types. While the information is useful, it would be better to include a table that summarizes these PDO biobanks, the cancer types, current applications of these PDO biobanks, and references.
Response 3: In response to this suggestion, we have included a comprehensive table summarizing their cancer types, applications, and key references. This table enhances the clarity and accessibility of information in Section 2.2, providing a succinct overview for readers and addressing the need for detailed PDO summaries across different cancer types.
Comment 4: In section 3.2, the authors made a nice schematic illustrating the advantages of PDO systems that incorporate stromal and immune cell types. However, the main text lacks substantive discussion. The authors should strengthen this section by reviewing the original publications that have successfully combined PDOs with these cell types, including their techniques, their findings, and their translational potential.
Response 4: We thank for the reviewer comments. On Section 3.2.3 we have address the need for a more substantive discussion on PDO systems incorporating stromal and immune cells. The section now includes examples of PDO co-culture systems using peripheral blood mononuclear cells (PBMCs) and tumor-infiltrating lymphocytes (TILs). Techniques for integrating immune components, such as air-liquid interface (ALI) systems and advanced microfluidic platforms, are discussed in detail. Findings from these systems demonstrate their ability to preserve immune-tumor interactions, including CD8+ T cell and natural killer (NK) cell activation, cytokine secretion, and immune checkpoint responses.
For instance, studies integrating PDOs with PBMCs have shown promising results in modeling patient-specific immune responses to immunotherapies. These platforms enable real-time evaluation of cytokine profiles, immune cell infiltration, and tumor cytotoxicity. The revised text also highlights the translational potential of these systems for optimizing immunotherapy regimens, including combination therapies involving iNKT cells.
Additionally, we have incorporated references to seminal studies, such as Jenkins et al. (2018), which demonstrate the feasibility of PDO-immune co-cultures in evaluating immunotherapy efficacy. These examples provide a robust foundation for discussing the integration of stromal and immune cells in PDO models, emphasizing their relevance to iNKT cell research.
Round 2
Reviewer 1 Report
Comments and Suggestions for Authors
Authors have significantly improved the manuscript.
Reviewer 4 Report
Comments and Suggestions for Authors
The authors have addressed all my comments and greatly improved the manuscript. By incorporating more relevant references and providing an in-depth discussion on the integration of PDOs with iNKT cells and other immune cells, the revised manuscript effectively highlights the primary objective of the review. This current version is now suitable for publication.